# The Main Protease of SARS-CoV-2 as a Target for Phytochemicals against Coronavirus

**DOI:** 10.3390/plants11141862

**Published:** 2022-07-17

**Authors:** Shaza S. Issa, Sofia V. Sokornova, Roman R. Zhidkin, Tatiana V. Matveeva

**Affiliations:** Department of Genetics and Biotechnology, St. Petersburg State University, 199034 St. Petersburg, Russia; st103070@student.spbu.ru (S.S.I.); s.sokornova@spbu.ru (S.V.S.); st085586@student.spbu.ru (R.R.Z.)

**Keywords:** SARS-CoV-2, COVID-19, main protease, phytochemicals, potential inhibitor, polyphenols

## Abstract

In late December 2019, the first cases of COVID-19 emerged as an outbreak in Wuhan, China that later spread vastly around the world, evolving into a pandemic and one of the worst global health crises in modern history. The causative agent was identified as severe acute respiratory syndrome coronavirus 2 (SARS-CoV-2). Although several vaccines were authorized for emergency use, constantly emerging new viral mutants and limited treatment options for COVID-19 drastically highlighted the need for developing an efficient treatment for this disease. One of the most important viral components to target for this purpose is the main protease of the coronavirus (Mpro). This enzyme is an excellent target for a potential drug, as it is essential for viral replication and has no closely related homologues in humans, making its inhibitors unlikely to be toxic. Our review describes a variety of approaches that could be applied in search of potential inhibitors among plant-derived compounds, including virtual in silico screening (a data-driven approach), which could be structure-based or fragment-guided, the classical approach of high-throughput screening, and antiviral activity cell-based assays. We will focus on several classes of compounds reported to be potential inhibitors of Mpro, including phenols and polyphenols, alkaloids, and terpenoids.

## 1. Introduction

In late December 2019, a viral pneumonia outbreak emerged in Wuhan, China caused by a new strain of coronavirus that was identified as SARS-CoV-2 (severe acute respiratory syndrome coronavirus 2) [1,2,3,4]. Soon after, the outbreak was declared a public health emergency of international concern by the WHO, and later in March it was declared a global pandemic, named COVID-19 (coronavirus disease 2019) [5,6]. The virus spread vastly all around the world, causing, to date, more than 500 million confirmed cases and millions of deaths in one of the worst global health crises in modern history [7]. Although many vaccines have been approved worldwide, so far there is still no treatment for COVID-19, and only supportive and preventive measures are being applied to reduce the disease’s complications [8,9,10]. Moreover, in trying to adapt to changing environments, the virus has developed a number of mutations that could strongly affect its transmissibility and infectivity. These mutations are also prone to increasing and spreading worldwide, due to natural selection [11,12,13,14]. Therefore, considering the constantly emerging viral mutants and the absence of approved, fully effective medications, there is an urgent need for developing an efficient treatment for COVID-19.

## 2. SARS-CoV-2 Structure and the Main Protease of SARS-CoV-2 as a Potential Protein Target

Similar to other viruses in the Coronaviridae family, SARS-CoV-2 has a single-stranded, positive-sense RNA (+ssRNA) genome of approximately 29 kb [15,16]. The viral RNA is composed of more than six open reading frames (ORFs), the first one of which (ORF1) serves as a template for producing two polyproteins essential for viral replication and transcription: pp1a and pp1ab [17,18]. These two polyproteins undergo extensive processing by the viral main protease (Mpro) and another protease known as papain-like protease (PLP), producing 16 nonstructural proteins (NSPs) [3,19]. The other ORFs encode at least four main structural proteins: the spike (S), membrane (M), envelope (E), and nucleocapsid (N) proteins [17,20,21].

Mpro, also known as 3-chymotrypsin-like protease (3CLpro), is a 33.8 kDa, three-domain cysteine protease, essential for proteolytic maturation and viral replication [9,18,22,23]. Mpro was found to be conserved among coronaviruses (CoVs), along with some common features of its substrates in different CoVs [18,24]. In addition to its vital role in the SARS-CoV-2 life cycle, the absence of any closely related human homologous Mpro makes it an ideal protein target for potential antiviral drugs, as its inhibitors are unlikely to be toxic in humans [25]. Furthermore, vaccines, as we have learned from previous viruses, can represent a selection pressure resulting in the evolution of novel resistant viral mutants, which again highlights SARS-CoV-2 Mpro as a good drug target, as it is less subject to such selection pressure caused by vaccines targeting the viral spike protein [26,27].

## 3. Standard Approaches That Could Be Applied in the Search for Mpro Potential Inhibitors

### 3.1. Virtual In Silico Screening: (The Data-Driven Approach)

Enabled by the development of bioinformatics tools along with eased access to protein databases, virtual screening has proven to be a fundamental tool in drug design and drug repurposing research [28,29]. In the virtual screening approach, automated molecular docking tools are usually used to predict the best possible variant for binding one molecule to another, considering the best orientation with the best binding affinity [30]. These tools enable the screening of large numbers of candidates against a specific studied target, at a very low cost [31]. Virtual screening is a data-driven approach that can be either target-based, where a library of candidate ligands is docked against the target and analyzed, or ligand-based, where a similarity search or a machine learning strategy can be applied [32,33,34,35].

Since the beginning of the COVID-19 pandemic, a large number of studies around the world have used this approach to search for potential inhibitors of SARS-CoV-2 [22,36,37,38]. Joshi R.S. et al. used this approach in their study conducted in 2020 to scan over 7000 compounds from different origins against SARS-CoV-2 Mpro [39]. Another study conducted by Tallei E.T. et al. in 2020 used this approach to evaluate the potency of plant-derived bioactive compounds against Mpro, resulting in the identification of pectolinarin, hesperidin, nabiximols, rhoifolin, and epigallocatechin gallate as potential antiviral phytochemicals [40]. Research by Tahir Ul Qamar et al. also resulted in the identification of 5,7,3′,4′-Tetrahydroxy-2′-(3,3-dimethylallyl) isoflavone, amaranthin, licoleafol, calceolarioside B, and methyl rosmarinate as potential inhibitors of the target, using this approach [41]. Khaerunnisa S. et al. extended the list with kaempferol, quercetin, luteolin-7-glucoside, demetoxycurcumine, naringenin, apigenin-7-glucoside, oleuropein, catechin, curcumin, zingerol, gingerol, and allicin [42]. Essential oils have also shown their effectiveness against SARS-CoV-2 Mpro in silico [43,44,45]. Therefore, using this approach, multiple natural compounds have been identified as strong binders of SARS-CoV-2 Mpro, and some of them were also identified as multi-target inhibitors that could be applied in COVID-19 management approaches [36,37,38,39,40,41,42,43,44,45].

### 3.2. The Classical Approach of High-Throughput Screening (HTS)

High-throughput screening (HTS) is a method for automated testing of thousands to millions of compounds for their biological activity against specific targets on model systems [46]. The development of robotics, laboratory equipment, laboratory methods, and software for the control of sample preparation, incubation, results detection, and data processing has allowed the HTS approach to be used to quickly search for lead compounds. Therefore, it is possible to quickly and inexpensively test large libraries of chemical compounds for their biochemical activity [47].

In practice, HTS is implemented in the form of a large number of miniature in vitro assays to identify molecules that can modulate the activity of a biological target. These reactions are run in 96-well, 386-well, or 1536-well plates [48]. Most often, the results of such biochemical analyses are obtained using various fluorescence detection methods [49], for example, direct measurement of fluorescence, fluorescence polarization, fluorescence resonance energy transfer (FRET), fluorescence quenching energy transfer (QFRET), or time-resolved fluorescence [46].

Since the early stage of COVID-19 pandemic, a large number of HTS assays have been developed worldwide to screen huge libraries of either previously approved drugs or potential inhibitors against SARS-CoV-2 [50,51,52,53]. Using this approach in their study to screen a library of 10,755 potential inhibitory compounds against SARS-CoV-2 Mpro, along with drugs previously approved for other viruses, Zhu W. et al. identified 23 potential inhibitors with different half-maximal inhibitory concentration values (IC_50_), the efficacy of 7 of which was confirmed in a later cytopathic effect assay [54]. Given the high safety level required of laboratories studying and manipulating the live SARS-CoV-2 virus (BSL-3 laboratories), Zhang, Q.Y. et al. proposed a new HTS assay to enable potential antiviral testing in a BSL-2 research facility, where they constructed a reporter replicon of the virus using *Renilla* luciferase (Rluc) reporter gene and validated it later using hit natural compounds [52]. Froggatt H. M. et al. also developed a fluorescence-based HTS assay using a protein derived from green fluorescent protein (GFP) to serve as a target for SARS-CoV-2 Mpro, and hence a reporter for the enzyme’s inhibition and activation, enabling rapid screening of libraries and identification of lead compounds [55]. A further improvement of HTS can be achieved by combining it with the previous in silico approach to yield an ultra-high-throughput virtual screening approach, where huge libraries can be tested against multiple viral targets efficiently and rapidly [56]. Gorgulla C. et al. conducted a study to search for SARS-CoV-2 inhibitors using this large-scale HTS screening approach and were able to screen over one billion candidate molecules against 40 different target sites on 17 potential targets, both in the virus and the host [56]. Although the results were obtained from computational data and have not all been tested with experimental analyses yet, this filtration of candidates could narrow down research targets for later more detailed and efficient analyses.

### 3.3. Antiviral Activity Cell-Based Assays

Cell-based assays offer an advantage over virtual or biochemical screening assays, as they provide a whole physiological environment, reflecting the complexity of a living system rather than focusing on a specific isolated target and thereby enabling a more accurate evaluation of the biological activity and potential toxicity of screened compounds [57,58,59]. Due to practical considerations, it is important to develop and test drug compounds that exhibit inhibitory activity at various stages of the virus life cycle. Therefore, test systems have been developed to evaluate the effectiveness of inhibitors of entry, uncoating, replication, assembly (in which viral proteases are active), and maturation of viruses [60]. However, the whole variety of such systems can be reduced to two main mechanisms for their implementation: cytopathic and reporter mechanisms [61]. In the first case, the activity of antiviral agents is assessed by reducing the formation of plaques due to the accumulation of coloring or luminescent agents in living cells [62,63]. In the second, viruses and cells with report inserts are used, and the activity of inhibitors helps in reducing the expression of the reporter protein [64].

Since work with a live virus is accompanied by significant organizational restrictions, approaches have been developed for evaluating the effectiveness of antiviral agents that model various stages of the life cycle of viruses in cells of HeLa [65], *Escherichia coli* [66], and *Saccharomyces cerevisiae* [61]. Moreover, cell-based assays are nowadays increasingly integrated into HTS assays to accomplish rapid screens in a relevant physiological environment [58].

Several recent studies have used antiviral activity cell-based assays, either after or combined with the previously described approaches, to investigate previously approved drugs and herbal medicines for their potential inhibition potency against SARS-CoV-2 [9,67,68,69,70]. Applying this methodology followed by a further in vivo validation, Jan J.T. et al. screened a 3000-candidate library of both pharmaceuticals and herbal medicines to test their effectiveness against SARS-CoV-2 Mpro and RNA polymerase and proposed multiple herbal extracts as potential herbal inhibitors against the targeted viral enzymes [68]. Another recent study conducted by Qiao J. et al. also applied this approach to investigate 32 different inhibitors against SARS-CoV-2 Mpro, 6 of which were found to have a high inhibition potency and were used to select candidates for further in vivo investigation [71].

The phased use of these three approaches makes it possible to identify those that exhibit the targeted therapeutic activity from the variety of known plant secondary metabolites. Among these compounds, there may be those that have not previously exhibited such properties. Furthermore, compounds for which hypothetical activity is found can be quickly tested for their effectiveness on cell-free and then on cellular systems. Such a screening strategy has shown to be effective in the search for inhibitors of SARS-CoV-2 Mpro, which indicates its potential in the search for drugs against new pathogens [9].

Therefore, at each stage of applying these methods, it is possible to significantly narrow the range of compounds under study, which facilitates a significant simplification and accelerates the search for molecules with clinical potential, therefore enabling moving on to the next stage of preclinical trials as soon as possible.

## 4. Phytochemicals as a Reservoir to Search for SARS-CoV-2 Mpro Potential Inhibitors

As mother nature has always provided an infinite library of natural products and chemicals, the use of herbal medicines and their derivatives to combat diseases dates back to more than 60,000 years ago in ancient history [72,73,74]. A study by Fabricant D. S. and Farnsworth N. R. in 2001 estimated that there were more than 250,000 species of higher plants on our planet, of which only 6% had been tested and evaluated for their biologic activity at the time [72]. Today, advanced screening techniques and assays have led to phytochemicals composing a significant part of the pharmaceutical market [75,76,77,78,79,80,81,82]. Plants can be used as sources of medicinal active compounds using several methodologies. In some cases, the whole plant or parts of the plant could be used as herbal remedy, e.g., garlic or curcumin [72,78,83]. Another methodology uses the plant as a direct source of bioactive compounds such as digoxin or morphine [72,84]. Sometimes, plants can provide compounds that could be used later as a starting point for producing highly effective, less toxic, easy-to-obtain, semisynthetic or synthetic compounds, as in the case of narcotic analgesics [72,85]. With such knowledge, thousands of studies all over the world have been conducted on searching for anti-SARS-CoV-2 treatments from plant origins, with Mpro being one of the most targeted viral components in this research, and the in silico data-driven approach being the most frequently applied [23,73,86,87].

Several classes of bioactive phytochemicals have been shown to be potential inhibitors of SARS-CoV-2 Mpro, including phenols [88,89], polyphenols [90,91], terpenoids [92], etc. (Figure 1). The main classes of SARS-CoV-2 Mpro inhibitors and their specific representatives are summarized in Table 1.

Most often, SARS-CoV-2 Mpro inhibitors are found among flavonoids and terpenoids. A common feature of compounds of these classes that exhibit SARS-CoV-2 Mpro inhibitory activity is an alpha-beta-unsaturated ketone group conjugated to an aromatic ring. It has also been shown that the presence of bicyclic aromatic rings in the structure and the presence of hydroxyl groups on all rings, especially on the B-ring of flavonoids, increase their respective activities [37,123].

Flavonoids, which are widespread in plants, are represented by compounds of various structures, often present together, which can lead to synergistic effects, including antiviral properties, for example, in tea, garlic, fruits, vegetables, etc. [124,125].

A study conducted by Nguyen T. et al. investigated the potential inhibitory effects of plant-derived polyphenols on SARS-CoV-2 Mpro, mainly those derived from black garlic extract prepared by heating raw garlic (*Allium sativum*) to high temperatures [91]. The studied extract contained many polyphenols, including both phenolic acids and flavonoids [126], of which several were found to have inhibitory effects against SARS-CoV-2 Mpro and were selected for further determination of their IC50 values (Table 1) [91].

*Salvadora persica* contains 10 flavonoid metabolites that were found to have substantially stable binding affinities for the SARS-CoV2 Mpro, including glycosides of kaempferol and its O-methylated derivatives [104]. Another study, conducted by Jo S. et al., used a fluorescence resonance energy transfer assay (FRET) to screen a flavonoids library against SARS-CoV2 Mpro, resulting in the identification of three flavonoids as potential inhibitors: herbacetin, rhoifolin, and pectolinarin [107]. Plants from Indian traditional medicine have also been found to contain potential inhibitors of SARS-CoV-2 Mpro [23,87]. One example is tulsi (*Ocimum sanctum*), with its derived polyphenols vicenin and isorientin 4′-O-glucoside 2″-O-p-hydroxybenzoate [87]. Another example of flavonoids derived from Indian medicinal plants showing such inhibitory effects was also presented in a recent study conducted by Saravanan K. et al. in 2020 [23]. In this study, 41 compounds from different plants were docked against SARS-CoV-2 Mpro. Several flavonoids of the candidate compounds showed relatively high binding affinity values, with the highest value being that of amentoflavone, a flavonoid derived from *Torreya nucifera* that has previously been shown to have in vitro antiviral properties [23,127].

Catechins are another group of phytochemicals representing a subclass of polyphenolic compounds found in a variety of plants and plant-derived dietary supplements such as green tea, cocoa, vinegar, wine, and garlic [91,128]. Due to the 3-galloyl and 5′-OH groups in their structure [129], catechins from green tea, mainly the previously mentioned epigallocatechin-3-O-gallate (EGCG), were found to exhibit antiviral properties against SARS-CoV-2 and specifically against its Mpro [88]. The same compounds were found in black garlic extract [91] and the petals of Himalayan *Rhododendron arboreum* [130], with similar potential anti-SARS-CoV-2-Mpro properties, using both in silico and in vitro analyses. In green tea (*Camellia sinensis*), three polyphenols were found to have good binding affinities for SARS-CoV-2 Mpro: EGCG, epicatechingallate (ECG), and gallocatechin-3-gallate (GCG). They were proven to be highly stable, similarly compacted, and subject to low conformational variability [110]. EGCG, found in different plant sources including *Camellia sinensis*, *Vitis vinifera*, and black garlic extract, was further investigated in several in vitro and in vivo studies to test its effectiveness against COVID-19 [131,132,133,134]. EGCG and its oxidized form were found to inhibit Mpro in vitro [133,134], to directly inhibit an early infection [132], and to reduce viral replication in mouse lung cells [132].

Lactones derived from plants have also shown potential inhibitory anti-SARS-CoV-2-Mpro properties [89]. An example of such lactones is withaferin A, a steroidal lactone derived from the well-known Indian medicinal plant Ashwagandha (*Withania somnifera*), which has shown inhibitory potencies against the targeted protease in a molecular docking study conducted by Sudeep H.V. et al. in 2020, with a binding score of −9.83 Kcal/mol [89].

In addition, plant tannins have been reported to have antibacterial, antifungal, and antiviral properties [135], and accordingly have been screened in several studies to investigate their potential inhibitory effects against SARS-CoV-2 Mpro [121]. A study by Wang S.C. et al. tested the effectiveness of tannic acid, a tannin found abundantly in red wine and in berries, grapes, pomegranate, and other fruits [136], against SARS-CoV-2 Mpro both in silico and in vitro, using pseudotyped viral particles. The obtained data suggested that tannic acid was a potential inhibitor of the targeted enzyme, as it was found to form a thermodynamically stable complex with Mpro. Cinnamtannin-B is another naturally occurring tannin that has been reported as a top hit against SARS-CoV-2 Mpro [137]. Cinnamtannin-B is derived from the cinnamon plant (*Cinnamomum zeylanicum*) and can only be isolated from a limited number of plants such as *Linderae umbellatae* and bay laurel (*Laurus nobilis*) [138].

Alkaloids represent another group of natural phytochemicals that have a broad spectrum of biological activities, mainly antiviral [139]. One example is capsaicin, a plant-derived alkaloid that is derived mainly from the fruit of the *Capsicum* genus [140] and has been found by in silico research to be a potential inhibitor of SARS-CoV-2 Mpro, similar to another plant-derived alkaloid named psychotrine [112]. Achyranthine is another alkaloid derived mainly from *Achyranthes aspera* [141] that was found to bind three sites of Mpro, with binding scores ranging between −4.1 and −4.7 [113].

Terpenes and their modified class of terpenoids represent a huge group of phytochemicals that have also been found to have antiviral properties in general and inhibitory potential effectiveness against SARS-CoV-2 Mpro in particular [92]. Plant terpenoids with medicinal potential are estimated to include more than 100,000 compounds on our planet, with more than 12,000 belonging to the diterpenoid group alone [142]. Of the ayurvedic medicinal plants, two terpenoids were suggested by in silico research as potential inhibitors of SARS-CoV-2 Mpro: ursolic acid from tulsi (*Ocimum sanctum*) and withanoside V from ashwagandha (*Withania somnifera*) [87]. Other examples of terpenes from Indian medicinal plants with inhibitory properties against SARS-CoV-2 are listed in Table 1. Some of these terpenes, e.g., curcumin, have already been proven to show antiviral activity in humans, protecting against acute and chronic lung diseases including pneumonia [143,144], and accordingly have been further suggested for application in clinical use as a prophylactic measure against COVID-19 [145]. From *Citrus limon*, two cyclic monoterpenes have been proven to interact with SARS-CoV-2 Mpro, including limonene and sabinene [113]. Glycyrrhizic acid (glycyrrhizin), another plant-derived triterpenoid saponin found mainly in *Glycyrrhiza glabra*, was also proven to have in vitro anti-SARS-CoV-2-Mpro potential [116]. Another main source of terpenes, mainly monoterpenes, is plant essential oils [146], which have been proven in several studies to have a wide variety of antimicrobial properties [147,148,149]. One example of such a plant-essential-oil-derived monoterpene with antiviral properties is pinene, a bioactive compound of black pepper (*Piper nigrum*), that was proven to bind Mpro in silico [113,150]. Similarly, the diterpenoid andrographolide from *Andrographis paniculate* was successfully docked against SARS-CoV-2 Mpro in two different molecular docking studies and was therefore suggested as a potential inhibitor to be evaluated in further in vitro analyses [89,151]. From medicinal Arabic plants, betulinic acid, a pentacyclic triterpenoid derived from the Christ’s thorn plant (*Ziziphus spina-christi*), was found by in silico research to successfully bind SARS-CoV-2 Mpro [152]. Roots of the plant *Maprounea africana* are also considered to be a source of several bioactive compounds, including triterpenes [153] such as 1β-hydroxyaleuritolic acid 3-p-hydroxybenzoate that have shown in silico inhibitory potential against Mpro [119].

## 5. Perspectives of Large-Scale Synthesis of Anti-SARS-CoV-2 Compounds

The large-scale production of newly discovered compounds with anti-SARS-CoV-2 activity can proceed in several ways. Some compounds can be obtained relatively easily by means of chemical synthesis. This applies to many of the flavonoids. Others are preferably obtained using cell cultures. A number of anti-SARS-CoV-2 compounds have attracted interest in the past due to their wide range of biological activities. For this reason, methods for their production in cell cultures have been developed and constantly optimized. For example, compounds such as tanshinones and rosmarinic acid can be effectively produced in hairy root cultures of *Salvia miltiorrhiza* Bunge [154] and catechins are produced in hairy roots of *Camellia sinensis* (L.) O. Kuntze [155].

In addition, there are several approaches for increasing the yield of specialized metabolites, including metabolic engineering of tanshinones [154,155,156], phenolic acids [155,157], flavonoids [158], and diterpenoids [159] or varying the cultivation conditions [155,157].

Finally, natural compounds can be precursors for subsequent chemical modification. For example, chemical synthesis based on chalcones allowed the development of more effective anti-SARS-CoV-2 compounds [160].

## 6. Conclusions

Plants have been used for a long time as a resource for bioactive compounds and phytochemicals to be applied in therapeutic approaches for different diseases. Since the COVID-19 pandemic is still ongoing, phytochemicals could be used to find effective and safe treatments for the disease. To date, using computer modeling of cell-free and cell-based screening approaches, some progress has been made in the search for potential drugs aimed at inhibiting the main protease of the coronavirus. They are represented by phytochemicals from several classes, including polyphenols, terpenoids, catechins, lactones, and tannins. Some plants containing promising compounds can be used as food directly, e.g., garlic, and others can serve as sources of pure substances for pharmacology. Future studies should shift their focus towards assessing possible toxic effects on cells, since even the most promising protease inhibitors will not be able to find application if they are found to have a toxic effect.

## Figures and Tables

**Figure 1 plants-11-01862-f001:**
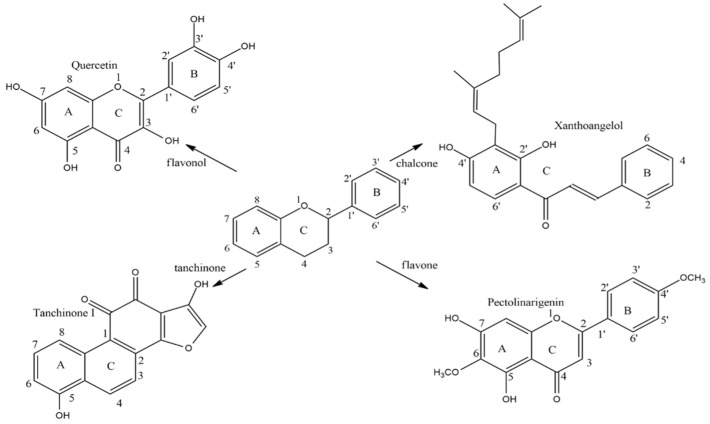
Basic flavonoid structures.

**Table 1 plants-11-01862-t001:** Phytochemicals reported to have potential inhibitory properties against SARS-CoV-2 Mpro.

Class of Compound	Compound	Distributed in	Including Food Plants and Spices *	Inhibition (%), with IC50 Value, μM	Binding Energy ** (kcal/mol)	Ref.
Isoflavone	Daidzein	Streptomyces, and predominantly in *Fabaceae* plant family	*Vigna radiata*, *Glycine max*	56	−6.5	[93]
Isoflavone	Puerarin	Predominantly in *Fabaceae* plant family	*Cicer arietinum*, *Glycine max*, *Glycyrrhiza glabra*, *Phaseolus vulgaris*, *Pisum sativum*, *Vigna radiata*	42 ± 2	−6.63	[94]
Flavonol	Myricetin	Widely distributed in various plant families	*Buchanania lanzan*, *Mangifera indica*, *Asparagus officinalis*, *Davidsonia pruriens*, *Hippophae rhamnoides*, *Vicia faba*, *Salvia hispanica*, *Thymus capitatus*, *Punica granatum*, *Hibiscus sabdariffa* L., *Moringa oleifera*, *Eugenia jambolana*, *Pimenta dioica*, *Plinia pinnata*, *Syzygium aromaticum*, *Syzygium cumini*, *Syzygium samarangense*, *Diploknema butyracea*, *Ampelopsis grossedentata*, *Morella rubra*	43 ± 1	−22.13	[41,91]
Flavonol	Quercetin	Mucor hiemalis, and widely distributed in various plant families	*Allium cepa*, *Allium Sativum*, *Allium ascalonicum*, *Mangifera indica*, *Annona muricata*, *Asparagus officinalis*, *Capparis spinosa*, *Carica papaya*, *Garcinia cowa*, *Garcinia dulcis*, *Brassica oleracea* var. *gongylodes*, *Raphanus sativus*, *Momordica charantia*, *Ceratonia siliqua*, *Vicia faba*, *Crocus sativus*, *Punica granatum*, *Toona sinensis*, *Moringa stenopetala*, *Musa acuminata*, *Psidium guajava*, *Phyllanthus emblica*, *Zea mays*, *Nigella sativa*, *Eriobotrya japonica*, *Prunus avium*, *Kadsura heteroclita*, *Capsicum annuum*, *Zingiber officinale*	93 ± 5	−7.6	[23,91]
Flavonol	Quercetagetin (Quercetagenin)	*Asteraceae*, *Eriocaulaceae*, *Fabaceae* plant families	*Acacia catechu*, *Leucaena glauca*	145 ± 6	−15.2	[91]
Flavanonol	Ampelopsin (dihydromyricetin)	Widely distributed in various plant families	*Asparagus officinalis*, *Punica granatum* L., *Pimenta dioica*, *Zea mays*, *Syzygium cumini*, *Capsicum annuum*, *Vitis rotundifolia*, *Manilkara zapota*,	128 ± 5	−7.5	[91,95]
Flavanonol	Ampelopsin-4′-O-α-d-glucopyranoside	Widely distributed in various plant families	*-*	195 ± 5	7.4	[91]
Flavanone	Naringenin	Widely distributed in various plant families	*Camellia sinensis*, *Prunus cerasus*, *Prunus persica*, *Citrus grandi*, etc.	150 ± 10	−7.7	[91,96]
Flavan-3-ol	Epigallocatechin gallate (EGCG)	Widely distributed in various plant families	*Vitis vinifera*	171 ± 5	−7.6 −8.2	[88,91,93]
Flavone	Vitexin	Widely distributed in various plant families	*Pisum sativum*	180 ± 6	−7.6	[97]
Hydrocinnamic acid	Chlorogenic acid	Widely distributed in various plant families	*Lactuca sativa*	39.48 ± 5.51	−12.98	[91,98]
Dihydroxycinnamic acid	Caffeic acid	Widely distributed in various plant families	*Actinidia deliciosa*, *Allium Sativum*, *Mangifera indica*, *Ilex paraguariensis*, *Carica papaya*, *Beta vulgaris*, *Terminalia catappa*, *Terminalia chebula*, *Brassica oleracea* var. *gongylodes*, *Raphanus sativus*, *Momordica charantia*, *Arachis hypogaea*, *Cicer arietinum*, *Glycine max*, *Phaseolus vulgaris*, *Pisum sativum*, *Tetrapleura tetraptera*, *Ocimum basilicum*, *Rosmarinus officinalis*, *Thymus capitatus*, *Punica granatum*, *Triticum aestivum*, *Zea mays*, *Crataegus pinnatifida*, *Prunus avium*, *Coffea arabica*, *Citrus limon*, *Citrus sinensis*, *Solanum lycopersicum*, *Solanum phureja*, *Solanum pimpinellifollium*, *Solanum tuberosum*, *Curcuma longa*, *Bergera koenigii*	197 ± 1	−12.4985	[91,99]
Polyphenol	Ellagic acid	Widely distributed in various plant families	*Mangifera indica*, *Terminalia chebula*, *Punica granatum*, *Moringa oleifera*, *Moringa peregrine*, *Moringa stenopetala*, *Eugenia jambolana*, *Myrciaria cauliflora*, *Syzygium aromaticum*, *Syzygium cumini*, *Emblica officinalis*, *Rubus idaeus* L.,	11.8 ± 5.7	−15.955	[100,101]
Phenylpropanoid	Chicoric acid	*Alliaceae*, *Asteraceae*, and *Labiatae* plant families	*Lactuca sativa*, *Ocimum basilicum*, *Cichorium intybus*	-	−8.2	[93]
Polyphenol	Gallocatechin gallate (GCG)	*Cistaceae*, *Elaeagnaceae*, *Ericaceae*, *Polygonaceae*, *Theaceae*, and *Vitaceae* plant families	*Hippophae rhamnoides*, *Camellia sinensis*, *Vitis vinifera*	5.774 ± 0.805	−9	[93]
Flavan-3-ol	Epicatechin gallate (ECG)	*Cistaceae*, *Elaeagnaceae*, *Ericaceae*, *Polygonaceae*, *Theaceae*, and *Vitaceae* plant families	*Hippophae rhamnoides*, *Camellia sinensis*, *Vitis vinifera*, *Rheum* sp.	12.5	−8.2	[93,102]
Flavonoids	Kaempferol glycosides	Widely distributed in various plant families	*Prunus avium*, *Allium cepa*	125.00	−7.4, −8.1	[103,104]
Flavonoids	Isorhamnetin glycosides	Widely distributed in various plant families	*Brassica oleracea*, *Allium ascalonicum*	13.13	−6.6, −8.2	[104,105]
Flavonoids	Pectolinarin	*Labiatae*, *Plantaginaceae*, and *Verbenaceae* plant families		37.7	−8.2	[40,106,107]
Flavonoid	Herbacetin	*Asteraceae*, *Chenopodiaceae*, *Crassulaceae*, *Ephedraceae*, *Equisetaceae*, *Linaceae*, *Malvaceae*, *Papaveraceae*, *Phrymaceae*, *Primulaceae*, *Rosaceae*, *Rutaceae*, and *Taxaceae* plant families	*Linum usitatissimum*, *Citrus limon*	33.1	−7.2	[40,106]
Flavonoid	Rhoifolin (apigenin-7-O-rhamnoglucoside)	*Acanthaceae*, *Anacardiaceae*, *Apocynaceae*, *Fabaceae*, *Lythraceae*, *Oleaceae*, *Rutaceae*, and *Caprifoliaceae* plant families	*Vicia faba*, *Hordeum vulgare* L	27.4	−8.2	[40,106]
Flavonoid metabolite	Vicenin	Rare compound	*Trigonella foenum-graecum*	38.856	−8.97	[87,108]
Flavone	Isorientin 4′-O-glucoside 2″-O-p-hydroxybenzoate	*Gentianaceae* and *Lamiaceae*	*Ocimum sanctum*	-	−8.55	[87]
Biflavonoid	Amentoflavone	Various plant species: yew, juniper, oak, and willow	*Garcinia brasiliensis*, *Garcinia dulcis*, *Garcinia hombroniana*, *Garcinia indica*, *Garcinia intermedia*, *Garcinia livingstonei*, *Garcinia madruno*, *Garcinia mangostana*, *Garcinia morella*, *Garcinia wightii*, *Garcinia xanthochymus*,	-	−10.0	[23]
Flavonoid	Silymarin (silibinin)	*Asteraceae*	*Silybum marianum*	46.88	−7.6	[23]
Isoflavone	Torvanol A	*Solanaceae*		-	−7.5	[23]
Flavone	Scutellarein	*Asphodelaceae*, *Asteraceae*, *Fabaceae*, *Bignoniaceae*, *Labiatae*, *Plantaginaceae*, *Polygonaceae*, and *Verbenaceae* plant families	*Oroxylum indicum*, *Garcinia andamanica* *Origanum majorana*	-	−7.4	[23]
Flavone	Apigenin	Widely distributed in various plant families, mainly in *Labiatae*	*Artemisia diffusa.*, *Ocimum americanum* var. *pilosum* *Ocimum basilicum*, *Ocimum x citriodorum*, *Rosmarinus officinalis*, *Salvia officinalis*, *Thymus piperella*, *Passiflora foetida*, *Piper peepuloides*, *Kaempferia parviflora*	925	−7.1	[23,109]
Isoflavone	5,7,3′,4′-Tetrahydroxy-2′-(3,3-dimethylallyl) isoflavone	Fabaceae		-	−29.57	[41]
Polyphenol	Methyl rosmarinate	*Labiatae*	*Rosmarinus officinalis*	21.32	−20.62	[41,110]
Flavonoid	Baicalin	Mainly in *Labiatae*	*Oroxylum indicum*	6.41 ± 0.95 μM in vitro 27.87 ± 0.04 μM in cells	−8.85	[99,111]
Flavonoid	Baicalein	Mainly in *Labiatae*	*Oroxylum indicum*	0.94 ± 0.20 μM in vitro 2.94 ± 1.19 μM in cells		[98]
Alkaloid	Capsaicin	*Solanaceae*	*Capsicum annuum*	-	−13.90	[112]
Alkaloid	Psychotrine	Mainly in *Rubiaceae*		-	−13.5	[112]
Alkaloid	Achyranthine	*Amaranthaceae*		-	4.1	[113]
Terpenoid	Withanoside V	*Solanaceae*		5.774 ± 0.805	−10.32	[114]
Triterpenoid	Ursolic acid	*Apocynaceae*, *Asteraceae*, *Boraginaceae*, *Dryopteridaceae*, *Ericaceae*, *Gesneriaceae*, *Labiatae*, *Lamiaceae*, *Lythraceae*, *Moraceae*, *Myricaceae*, *Nothofagaceae*, *Oleaceae*, *Rosaceae*, *Rubiaceae*, *Solanaceae*, *Stilbaceae*, and *Ulmaceae* plant families	*Vaccinium macrocarpon*, *Punica granatum*, *Olea europaea*, *Prunus avium*, *Pyrus* spp.	12.6	−8.2	[114]
Triterpenoid	Glycyrrhizic acid	*Asteraceae* and *Fabaceae*	*Stevia rebaudiana*, *Glycyrrhiza glabra*	-	−8.03	[115,116]
Pentacyclic triterpenoid	Torvoside H	*Solanaceae*	*-*	-	−8.4	[23]
Pentacyclic triterpenoid	Lupeol	In *Coprinaceae*, and widely distributed in various plant families	*Cichorium intybus*, *Zanthoxylum armatum*, *Olea europaea*, *Myrica rubra*, *Morus alba*, *Ficus carica*, *Carica papaya*	-	−7.6	[116]
Diterpene	Scopadulcic acid B	Plantaginaceae (Scoparia dulcis)	*-*	-	−8.5	[23]
Diterpene	Ovatodiolide	*Lamiaceae*	*-*	-	−6.9	[23]
Terpene	Curcumin	Mainly in *Zingiberaceae*	*Curcuma longa*, *Curcuma mangga*, *Zingiber officinale*	11.9	−6.5	[23,117]
Terpene	Parthenolide	*Asteraceae*, *Magnoliaceae*, and *Celastraceae* plant families	-	-	−6.0	[23]
Meroterpenoid	Illicinone A	*Illiaceae*	*Illiciumverum*	-	−5.0	[23]
Meroterpenoid	Piperitenone	*Labiatae* and *Poaceae*	*Mentha* spp.	-	−4.3	[23]
Cyclic monoterpene	Limonene	Widely distributed in various plant families	*Citrus aurantium*, *Citrus aurantifolia*, *Citrus bergamia*, *Citrus grandis*, *Citrus junos*, *Citrus latifolia*, *Citrus limettioides*, *Citrus limon*, *Citrus medica*, *Citrus paradisi*, *Citrus reticulata*, *Citrus sinensis*, *Zanthoxylum armatum*, *Allium sativum*, *Anacardium occidentale*, *Mangifera indica*, *Monodora myristica*, *Xylopia aethiopica*, *Cuminum cyminum.* *Foeniculum vulgare*, *Petroselinum crispum*, *Porophyllum ruderale*, *Beta vulgaris*, *Ocimum basilicum*, *Thymus piperella*, *Acca sellowiana*, *Psidium guajava*, *Averrhoa carambola*, *Piper nigrum* *Prunus avium*, *Coffea arabica*, *Coffea canephora*, *Citrus aurantifolia*, *Curcuma amada*, *Curcuma longa*	-	−5.2	[113]
Cyclic monoterpene	Sabinene	*Labiatae*, *Cupressaceae*, *Myristicaceae*, and *Pinaceae* plant families	*-*	-	−4.8	[113]
Bicyclic monoterpene	Pinene	Widely distributed in various plant families	*Piper nigrum*, *Allium sativum*, *Anacardium occidentale*, *Mangifera indica*, *Pistacia vera*, *Monodora myristica*, *Cuminum cyminum*, *Foeniculum vulgare*, *Petroselinum crispum*, *Ocimum basilicum*, *Origanum vulgare*, *Rosmarinus officinalis*, *Myristica fragrans*, *Eriobotrya japonica*, *Fragaria vesca*, *Citrofortunella mitis*, *Citrus aurantifolia*, *Citrus* spp., *Curcuma mangga*, *Curcuma amada*, *Aframomum melegueta*, *Solanum lycopersicum*, *Zanthoxylum armatum*,	-	−4.6	[113]
Labdane diterpenoid	Andrographolide	*Acanthaceae*	*-*	-	−6.6	[118]
Triterpene	1β-hydroxyaleuritolic acid 3-p-hydroxybenzoate	*Euphorbiaceae*	*-*	-	−8.5	[119]
Steroidal lactone	Withaferin A	*Solanaceae*	*-*	-	−9.83	[89]
Tannin	Tannic acid	*Ephedraceae* and *Geraniaceae*	*-*	13.4	−7.5	[120,121]

* KNApSAcK Core System (KNApSAcK DB group (skanayagtc.naist.jp). ** The binding energy of Mpro to Lopinavir (−9.1 Kcal/mol) or Nelfinavir (−8.4 Kcal/mol) is given for comparison of this value for herbal compounds [122].

## Data Availability

Not applicable.

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
