# Peer review of "The Main Protease of SARS-CoV-2 as a Target for Phytochemicals against Coronavirus"

_plants, 2022, doi:10.3390/plants11141862_

Round 1

Reviewer 1 Report

The paper “The main protease of SARS-CoV-2 as a target for phytochemicals against coronavirus” by Issa et al properly presents the phytochemical compounds and the approaches in use for the evaluation of the anti-SARS-CoV-2 effects. The authors presented the available data that indicates the inhibition of the coronavirus Mpro protease by phytochemicals.

The paper is well structured and well-written, with almost no editing mistakes. However, I have several suggestions for the authors.

1.      Even though, in the abstract, the authors state that they will focus on phenols, polyphenols, alkaloids, and terpenoids, they fail to cover the alkaloids and terpenoids. No example is given in Table I and the manuscript text. As an example, glycyrrhizin has been shown to inhibit the Mpro protease. I suggest including at least the main components that have been shown to have antiviral activities

2.      In addition to the presented data, I think it would be beneficial for this paper to include, if available (at least for some phytochemicals such as curcumin and EGCG there is data) clinical and animal-derived data to show that phytochemicals do have a positive effect on the treatment of this viral infection.

Based on these observations,  I recommend for this article to be accepted after minor revision.

Author Response

We would like to thank the reviewer for the thoughtful comments and efforts towards improving our manuscript. In the following, we highlight our revisions point by point in order to address the concerns raised by reviewer.

  • The alkaloids, terpenoids (including glycyrrhizin), and detailed examples of flavonoids were added in text (as marked up in pages 21, 22 and 23) and in table (from page 13 to 20).
  • Regarding in vivo/clinical studies on some phytochemicals we added information about EGCG and curcumin to pages 15 and 17, respectively.

Reviewer 2 Report

The review entitled: „The main protease of SARS-CoV-2 as a target for phytochemicals against coronavirus„  is of scientific interest in the context of the COVID 19 pandemic and the increased interest in alternative therapies.

Some suggestions and editing corrections are recommended:

·        The authors 'affiliation and the abbreviations of their names are not included in the authors' details as required the journal instructions.

·        Chapters 4 and 5 can be combined into one, referring to the same topic

·        Table 1 to be placed landscape

·        language and grammar corrections are required

Author Response

We would like to thank the reviewer for the thoughtful comments and efforts towards improving our manuscript. In the following, we highlight our revisions point by point in order to address the concerns raised by reviewer.

  • Authors names and affiliations were corrected
  • Section 5 was  merged with section 4
  • Table 1 was converted to landscape layout as suggested

Reviewer 3 Report

My comments are as below:
Abstract
Abstract is carelessly written authors should incorporate their notable findings and adequately connect with the sentences they choose to correspond.
Introduction
The introduction section must have a clear hypothesis and significantly develop the second paragraph of your manuscript. Make it more connecting to the problem statement.
Overall there is the repetition of the information, which could be avoided.
Discussion
This section should include more information and references related to the relevant and related works.

Conclusions
If possible, restructure and carefully edit the conclusion section and add clear information regarding the most noteworthy findings.

Author Response

We would like to thank the reviewer for the  comments 

Regarding the abstract, we addressed the issue of inconsistency between abstract and manuscript text, as advised also by the first reviewer, by adding more detailed examples of flavonoids, and also alkaloids and terpenes.

But for the other comments of the third reviewer, we would kindly like to ask if there is possibly a mix up in the review, as we did not find the comments apply to our review manuscript, but rather to an original research article or case report.